# Development and Implementation of a Novel Approach to Scaling the Meeting Centre Intervention for People Living with Dementia and Their Unpaid Carers, Using an Adapted Version of the Template for Intervention Description and Replication (TIDieR) Checklist

**DOI:** 10.3390/bs15050670

**Published:** 2025-05-14

**Authors:** Nathan Stephens, Shirley Evans, Christopher Russell, Dawn Brooker

**Affiliations:** Association for Dementia Studies, University of Worcester, Worcester WR2 6AJ, UK; shirley.evans@worc.ac.uk (S.E.); c.russell@worc.ac.uk (C.R.); dawn.brooker@worc.ac.uk (D.B.)

**Keywords:** scaling, meeting centre, psychosocial intervention, intervention development, intervention implementation, dementia, post-diagnostic support

## Abstract

Complex interventions are often poorly described, making it difficult to understand their development, implementation, and evaluation (i.e., context), which can aid in replicating and translating evidence into practice and policy. Therefore, we examined the development and implementation of an approach to scaling-out (increasing the numbers of) a psychosocial intervention for people affected by dementia across a rural county in England during 2019–2024. We used an adapted version of the TIDieR checklist to consider key items essential for replicating complex interventions at scale. We triangulated document analysis with interview data, and key programme stakeholders ratified the results of this study. We identified three programme components and sub-components, including how planned components were delivered, by whom, and under what conditions. Implementation outcomes related to the inner (e.g., top-down structure) and outer contexts (e.g., market failures) led to modifications that increased programme complexity. This study highlights the importance of programme leaders who can convene and drive systems and culture change to address implementation challenges, as well as the need for scaling science during programme development, for example, to determine optimal scale. Further research should include testing implementation outcomes to understand if the intervention is a scalable solution to the gap in post-diagnostic support in the UK.

## 1. Introduction

### 1.1. The Scaling Challenge for Psychosocial Interventions in Dementia

Psychosocial interventions in dementia should ideally move from proof of concept, looking at what worked (or not) in the real world, to informing scaling through health service integration and institutionalisation ([85]). Scaling is a systematic effort to alter the scope and impact of an intervention to better match the needs of populations ([111]; [112]). This definition includes variations: scaling up (impacting law and policy), out (impacting more people and places), deep (impacting societal culture and values), down (fitting to the local context), and in (altering organisational practices) ([68]). The implementation gap between successful pilot and routine adoption means there is limited availability of post-diagnostic support (PDS) intervention in the UK, and the existing services and support are fragmented and complex to navigate ([3], [4]). This leaves people affected by dementia on the margins of support and society ([110]), particularly those in highly deprived areas where health and wealth inequities are most acute ([109]). The gap in PDS exists despite the scaling-out of some ‘successfully tested’ dementia interventions being cost-effective ([52]). However, it is a fallacy to assume that establishing effectiveness or cost-effectiveness will lead to routine application. Improvement in adult social care is a complex and highly political process ([39]). Nevertheless, the concept of scaling is now mainstream in adult social care policy in England ([29]); however, as a practice and science, scaling lacks a developed evidence base ([23]). Therefore, it is important and timely that we develop evidence to understand and increase the spread, scale, and sustainability of good practice in areas of adult social care, including psychosocial interventions in dementia ([7]).

### 1.2. Increasing the Impact of Meeting Centres Through Scaling-Out

Framed in this context is a novel approach to scaling-out Meeting Centres (MCs) across a rural county of England: the Worcestershire Meeting Centres Programme (WMCP). The MC is “a local resource, operating out of ordinary community buildings, that offers ongoing … support to people with mild to moderate dementia and their families. At the heart of an MC is a social club where people…get help that focuses on what they need … to cope well in adjusting to living with” dementia ([34]). MCs are a psychosocial intervention that aim to empower people affected by dementia to accept their situation and engage with social (e.g., maintain social networks), emotional (e.g., positive self-image), and practical (e.g., professional relationships) support ([15]). Underpinning an MC is a psychological theory on coping and crisis, modelled in the UK as the process of Adjusting to Change, as shown in Figure 1 ([15]).

The MC ‘Essential Features’ (EFs) have been described ([34]), adaptive implementation modelled ([60]), and comparable effectiveness evaluated ([16]; [35]) in the UK, Italy, and Poland, providing ‘proof of concept’ (i.e., what it is and whether it work) and ‘real-world implementation’ (i.e., approach to and factors affecting implementation) ([85]). However, evidence to inform approaches to scaling-out MCs is lacking. This is despite MCs being recognised in national policy ([95]) and replicated over 60 times in the UK through a dissemination project, Meeting Centres UK (2018–2021) [MCUK]. The MCUK project raised awareness and advocated to a range of stakeholders, including grassroots organisations, national charities, and policymakers (dissemination), for the adoption and institutionalisation of MCs (diffusion) ([93]). MCUK has been a highly effective mechanism for supporting providers in setting up and sustaining an MC. However, the diffusion of MCs into routine policy and practice has been spontaneous, driven by individual actors in isolated locations, rather than a strategic effort driven by collective groups of individuals and agencies across multiple locations and system levels. Consequently, MCs are operational in many regions of the UK, yet are only available to a few people due to limited coverage. The WMCP is novel in this respect, targeting multiple implementation levels to affect change in different contexts across a population (i.e., county).

### 1.3. The Case for Intervention Description When Reporting Complex Social Interventions

The Medical Research Council recommends that programmes be described separately in the public domain to clarify key information for replication ([98]). This type of reporting has been overlooked due to reporting biases (e.g., the dominance of effectiveness studies) and insufficient space in articles ([20]; [32]; [46], [47]). However, adequately describing a health or care intervention supports an assessment of the circumstances under which it is developed, implemented, and evaluated (i.e., context) ([2]; [10]; [68]), which can help in understanding the factors affecting individual, organisational, and implementation outcomes ([91], [89]). Therefore, it is concerning that dementia interventions are seldom described using reporting guidelines registered on the EQUATOR Network, a platform established explicitly for Enhancing the QUAlity and Transparency Of health Research ([33]).

Without a systematic and comprehensive approach to describing WMCP development and implementation, there is limited to no information on the scaling strategy, underpinning the theory, key components, how these were implemented and adapted to real-world contexts, or why the programme was funded and developed in the first place ([24]; [31]; [40]; [47]). Researchers and practitioners need this information to understand context ([98]) and implement findings ([20]). We have considered the benefits of describing the WMCP systematically rather than briefly documenting it in the introduction or background of the work and decided the latter approach will not suffice if, as others acknowledge, the aim is for beneficial dementia interventions to reach large-scale adoption more quickly ([41]; [52]; [56]; [81]; [96]; [119]).

### 1.4. Research Question and Aims

To understand ‘How the WMCP was developed and implemented, by whom, and under what circumstances?’, we aimed to systematically describe the WMCP using an adapted version of The Template for Intervention Description and Replication (TIDieR) checklist to increase the translation of knowledge into practice and policy ([47]). TIDieR is described in the following Methods Section.

## 2. Methods

This study was part of a pragmatic evaluation of the WMCP carried out by the lead author of this paper (NS), reported separately ([99]). The evaluation used an exploratory sequential multi-level mixed methods design ([26]; [80]), consisting of four interlinked studies that described the WMCP development and implementation, constructed an initial programme theory, measured outcomes, and analysed costs and benefits (Figure 2). Underpinning the evaluation were principles of pragmatic evaluation approaches, methodological comprehensiveness and operational practicality ([25]). These principles manifest in the research design through the prudent choice of qualitative methods considered feasible to answer a proximate question focused on the development and behaviour of a complex adaptive system, the WMCP. We used TIDieR as a framework to critically describe the WMCP development and implementation ([47]).

TIDieR consists of reporting items to describe interventions in the public domain systematically and has been used to enhance the quality of reporting on clinical trials and applied health research ([2]; [10]; [31]; [87]). However, TIDieR was adapted in this study to include reporting items that could account for intervention complexity ([24]) and the science and practice of scaling ([40]). Table 1 shows the adapted TIDieR checklist used to describe the WMCP, including each reporting item, and where reporting guidance was adopted for each item (indicated by an asterisk*). The following explains how and why TIDieR was adapted to describe the WMCP.

Additional items for TIDieR when reporting complex interventions were incorporated to consider (i) the stage of implementation, (ii) how well the programme was delivered, and (iii) which stakeholders had been involved in completing TIDieR ([24]). We assumed that including these items/considerations would improve our ability to capture the complexity inherent in the WMCP development and implementation ([24]). For example, considering ‘how well’ the WMCP distributed the funding against how it planned to distribute it highlighted the influence of external contexts (e.g., how the market is shaped) on programme implementation and outcomes (e.g., equity, sustainability, fidelity). Accounting for complexity in the WMCP design and delivery was important because we took a complex systems perspective of MCs being nested within the WMCP (supra-system). This perspective acknowledged previous research that characterised MCs as ‘system innovations’ because their development and implementation involve and are influenced by factors at the micro/service delivery level, meso/organisational level, and macro/care system level ([63], [62]).

In addition, we considered the Standards for Reporting Trials Assessing the Impact of Scaling Up Strategies of Evidence-Based Practices (SUCCEED) ([40]), which is registered with EQUATOR as ‘under development’ ([33]). Indicative items from the SUCCEED protocol paper enabled us to report details specific to scaling programmes ([40]). For example, was the scalability assessed, was the optimal scale considered, and was implementation science theory prominent in the design and delivery of the WMCP? We needed to include specific items that could capture the different types ([68]), models ([65]), and principles ([53]) used in scaling science and practice, as these make it a distinct science and practice from implementation ([54]). However, SUCCEED was under development when this paper was submitted, and so the items used were not considered exhaustive.

### 2.1. Data Collection

This study was primarily a descriptive document analysis in the manner described by [13] ([13]). This method of collecting and evaluating different types of documents is an effective way to explore the implementation of complex community-based interventions ([92]; [94]). In the case of the WMCP, document analysis was justified because the programme development and implementation relied on emerging bureaucratic processes in which physical and electronic documents were developed, exchanged, and received by and between stakeholders (e.g., applications, annual reporting, publicity materials). [5] ([5]) and [13] ([13]) described these documents as “social facts” withholding knowledge that can help unpack the black box of programme implementation ([45]). Therefore, the inclusion of documents as a dataset was necessary to capture and understand why and how the WMCP interacted with its contexts by tracking the development and implementation through documents that, when combined with other methods as a means of triangulation, helped confirm, challenge, and complement the findings ([92]).

WMCP planning began prior to the research and continued throughout; therefore, document analysis was retrospective and prospective and included pre-implementation planning documents (n = 1), WMCP assessment panel meeting notes (n = 9), MC annual financial reporting (n = 6), MC fidelity assessments (n = 3), provider applications (n = 6), implementation resources (e.g., guidance for setting up an MC, data collection materials) (n = 3), and publicity and marketing materials (e.g., press releases, MC open days) (n = 8). The lead author (NS) collected documents from a secure OneDrive folder stored and managed by the WMCP Leader—University of Worcester, UK. However, the document analysis procedures did not include a detailed audit list to identify the purpose, creator, and exact information period. Therefore, it was difficult to determine the authenticity and accuracy of the selected documents. To improve trustworthiness ([19]), different stakeholder interview data (n = 34) were considered and triangulated with descriptive reporting of key programme information found in the documents. Please refer to Appendix B for a brief description of the interview method and participants.

### 2.2. Data Analysis

One researcher (NS) deductively analysed documents based on the reporting framework listed in Table 1 ([36]). Drawing on the work of the Scaling Community of Practice ([53]), stakeholders were categorised as either Funder (contributes to the programme delivery costs), Leader (manages programme design and day-to-day delivery), Intermediary (recognised role supporting programme delivery), or Provider (directly provides MCs).

The analytical process was iterative, involving superficial and thorough line-by-line examination of documents to identify relevant information, akin to qualitative content analysis ([13]). Despite being a primarily descriptive study, the analysis included the interpretation of data where the meaning was not explicit or required contextualisation in the context of the WMCP. For example, modifications reported later in this paper were distinguished by what implementation outcomes they attempted to address, despite never being implied as such in the documentation and only implicitly so during interviews. NS analysed the interview data thematically using implementation science theory ([91]) to construct the WMCP programme theory (reported separately). We used the interview data in this study to substantiate descriptive reporting. We included the accounts from practice because professional insight helps understand ‘what works’ in adult social care ([103]), adding a critical voice to the descriptive statements about what, how, and why the programme was developed and implemented, for example, the rationale and results of modifications.

A reflexive approach to analysing the data was necessary to identify relevant information and contextualise varied types and quality of documentary evidence. This meant drawing on NS’s sustained engagement since February 2021 with individual MCs, WMCP management, and the wider MCUK network. For example, NS had a role or had been present during the development of documents, the discussion of documents in programme meetings, and/or the exchange of documents through emails. Not all the documents were included in this study for ethical reasons alluded to below, but this ethnographic-like exposure to key programme-related content and activity was central to the sense-making process that qualitative document analysis requires ([58]; [92]; [114]).

A final layer of rigour was added to the analysis through member checking, enhancing the trustworthiness and validity of the results ([11]). We asked stakeholders who had a lead role in developing and implementing the WMCP to review and ratify the information used and the results of this study, including the WMCP manager responsible for the programme’s oversight and delivery, the financial administrator responsible for managing funding awards to providers and financial reporting to the Funder, and the research assistant who organised the MC data collection and WMCP evaluation. In addition to evaluating that the content used to describe the programme was appropriate ([13]), stakeholders clarified key components and sub-components of the WMCP, how implementation deviated from initial planning, and the factors affecting implementation outcomes (e.g., fidelity, reach) ([40]). Their involvement resulted in a more precise and reasoned depiction of the WMCP development and implementation.

### 2.3. Ethical Considerations

This research was granted ethical approval from the University of Worcester (REP CODE: CHLES21220001-R1). We adopted a systematic five-step recruitment process to establish individuals’ capacity and gain informed consent to participate in interviews, as described elsewhere ([72]). Noteworthy aspects of this process were the role of gatekeepers, the provision of ‘simple’ and ‘accessible’ information ([28]), and the determination that the reassessment of capacity always began with the assumption that people could make informed decisions ([64]).

Consent was required from the individual or organisation where documents included personal or sensitive data. To reduce the ethical risk, we did not include direct individual quotes (e.g., personal statements from people at MCs shown on MC marketing materials) and considered the cultural and historical contexts of findings (e.g., previous examples of failed provider relationships). Furthermore, consent was not necessary where publicly available documents were used, such as the Guidebook for setting up an MC ([17]); however, we cited the sources of these documents correctly and consulted the host organisation (e.g., the member-checking process described earlier). We only included digitally available documents, and no physical copies were made of documents so that information could be stored and managed more securely via the University of Worcester’s secure OneDrive service. Finally, information that could identify individuals was removed and/or anonymised as far as possible. However, anonymity could not be guaranteed because we viewed individual job roles as necessary reporting criteria for the findings to make sense and be useful in practice.

## 3. Results

The following two sections organise the TIDieR items according to the two dimensions of the research question: ‘How was the WMCP (1) *developed*, and (2) *implemented*?’ This structure provides a more streamlined report of the key findings than reporting each TIDieR item separately.

### 3.1. How Was the WMCP Developed

#### 3.1.1. Why: Rationale, Theory, and/or Objectives of the Programme

Following the sustainability of a pilot MC in Worcestershire, England, since 2015 and the capacity added by the MCUK project in 2018, the WMCP was launched in 2019 by the Worcestershire County Council [Funder] to replicate MCs county-wide (scaling-out). The programme provided GBP 540,000 from a Business Rates Retention Scheme introduced in England in 2013 ([73]). This scheme enabled the Council to retain up to 50% of any real-terms changes in business rates revenues to increase their spending power in local areas, providing additional means for Councils to fund programmes that meet local needs ([73]). The Association for Dementia Studies at the University of Worcester [Leader] led this implementation project. Awards of up to GBP 60,000 were available for third-sector organisations in three instalments (once per annum) towards the implementation costs. The Funder made a small contribution to the Leader for the programme’s management and evaluation. Table 2 demonstrates the WMCP core and sub-components. The programme’s ability to replicate the success of the pilot MC (scalability) was not assessed ([66]; [118]).

#### 3.1.2. The ‘Strategy’ to Scale-Out the Provision of the Meeting Centres

The development of the WMCP strategy recognises that relevant frameworks and principles should inform the implementation and evaluation of scaling programmes ([53]; [61]). The Funder implemented a seed-funding approach to influence the scaling-out of MCs (impacting more significant numbers) on the basis that the organisation delivering an MC (Provider) could secure investment from elsewhere to stimulate growth and sustainability ([117]).

The WMCP was considered a scaling-out approach based on the five typologies of scaling defined by [68] ([68]), which distinguishes scaling-out as replicating a successful social innovation in different communities/settings/locations to reach a more significant number of people. A focus on maximal scale drove the WMCP’s scaling-out approach, defined loosely by the programme’s primary ‘performance target’, as up to nine MCs within Worcestershire, with at least one in each of its Primary Care Networks and six districts. The WMCP intended to distribute funding to a range of third-sector providers. Other implementation objectives are demonstrated in Table 2 and discussed in the next section.

Modelling by the Funder forecast that the WMCP could reach ‘600 people based on 60 people attending 10 MCs on a given day and deliver a return on investment of GBP 1,830,114 through the avoided cost of residential care’ (GBP 915,057 for the Local Government). The forecast included the initial investment as a cost and avoidance of residential care as a benefit. Furthermore, there was no evidence of a preliminary situational analysis to explore internal and external factors affecting WMCP implementation, such as public preference, system/professional readiness, and supply/demand issues. Consequently, there was limited evidence to indicate the optimal scale based on the specific needs of target populations/areas, the nature of the programme, and environmental complexities ([53]).

At the service delivery level, evidence-based resources promoted implementation fidelity through ‘Essential Features’ (EFs) ([106]; [34]) and adaptive implementation to local contexts ([60]). These can enable providers to understand and balance the features needed to support fidelity while modifying them to be effective across different contexts when scaling-out ([88]). Performance targets were set for Providers by the Leader in line with the EFs (see Table 2).

The implementation structure of the WMCP also mattered as a strategic feature of the programme. The Leader established the Assessment Panel to assess applications and overall WMCP monitoring, including the Leader (programme manager, research assistant, and administrator), independent chair, and Local Government and Public Health representative. This ‘top-down’ approach meant that strategic decision-making by the Assessment Panel was independent of local-level professional and public opinion, experience, and context (‘bottom up’) ([88]).

### 3.2. How Was the WMCP Implemented

#### 3.2.1. What and Why Materials and Procedures Were Implemented

To promote the WMCP, the Leader published press releases, maintained social media activity, and held information sessions to identify applicants. Following up with prospective applicants via email and phone was important for building commitment to transition from registering an interest to applying. All potential applicants received implementation guidance ([17]) and application forms (online and physical). At a service delivery level, MC promotion and marketing resources were developed and distributed by Providers remotely (email and online) and in-person (posters and leaflets) to GP practices, PDS services, and relevant community settings. Intermediaries emailed key programme information and resources to relevant networks such as the Dementia Action Alliance and adult social care. Providers developed materials to increase MC uptake (e.g., advertising taster/trial days), and the Leader focused on material to penetrate key services and systems to increase referrals into MCs from primary and secondary care services (e.g., MC referral form).

Applications for the WMCP were made in five funding rounds to enable the piloting of the process and management of applications while continuing to offer technical support to applicants. It also shortened decision-making at Assessment Panel meetings due to the limited number of applications per round. The initial screening and scoring of applications were based on the EFs and completed by the Leader (research assistant). The Assessment Panel decided the outcome of applications as either awarded, not awarded, or awarded funding with amendments to the application.

MC staff and volunteers were encouraged to complete a free 5-week online course delivered by the Leader ([107]). This training included the EFs, the Adjusting to Change model, and the importance of involving/engaging families and professionals. The course ran three times a year throughout the WMCP implementation via an online teaching platform. Learning modalities included independent study materials (videos, resources, and literature) and a one-hour live-taught session each week by a lecturer ([107]). Attendance in (at least) four online sessions was needed to complete the course. In addition, Providers were sent data collection resources to collect and submit data routinely and were expected to attend an online data collection information session provided by the Leader. Routine data included attendance (reported monthly), satisfaction (every 6 months), and wellbeing measures for loneliness ([49]), quality of life ([43]), and emotional wellbeing ([100]) (baseline and every 6 months of follow-up). Measures collected by MC staff from people living with dementia and unpaid carers at MCs were sent to the Leader to monitor and report the reach and impact of the WMCP. Ethical approval granted permission to use the data for research purposes, which included a data-sharing agreement between each Provider and the Leader.

To assess fidelity, the Leader visited MCs at 9, 18, and 27 months to assess their adherence to the EFs. Where visits were not possible, reviews were conducted online between the Leader and Providers. The Leader provided a report every 6 months on the programme’s progress against targets set by the Funder, including coverage, unplanned changes (underspend/overspend, location changes), financial targets, risks, and mitigating actions. The Leader also reported to the Assessment Panel annually to approve/disapprove Providers’ payments based on whether they had been adhering to or working towards the EFs. Formative programme monitoring was implemented through monthly internal WMCP and quarterly Assessment Panel meetings. Topics included coverage, uptake, and data collection. MCs collected routine data concerning the impact on individual wellbeing and the effectiveness of the county-wide funding model. Summative and formative reporting findings were fed back to the Funder, Leader, and key intermediaries to support continuous learning cycles throughout implementation (feedback loops) ([83]).

#### 3.2.2. How Many, When, and Where Did Interventions Take Place During WMCP Implementation

The WMCP established 10 new MCs in Worcestershire. It took three to eight months for the MCs to open from the point that Providers were awarded funding. Providers planned to increase the number of days they opened weekly from 1 to 3 days in the final year. Similar year-on-year targets were set for reaching people with dementia and unpaid carers. To demonstrate where interventions took place, the WMCP was modelled (Figure 3). The model includes key interactions within and between levels, stakeholders, and the WMCP, including the direction of involvement ([25]). MC11 was the initial pilot MC operating prior to the WMCP. To complement this model, Appendix A reports the stakeholders involved in the WMCP and how they intended to support the programme.

At the macro level, national dementia policies provide the rationale for reducing the PDS gap ([30]; [95]). Through these policies, service/support integration and reduced inequality in health reforms are progressed ([75]) along with legislation ([42]). Health and social care workforce strategies contributed to understanding the programme’s value through recruiting, training, and retaining staff and volunteers ([97]) and filling the existing skills gap ([57]). The MCUK Community of Learning and Practice [CoLP], facilitated by the Leader, meets online fortnightly and includes community-based organisations nationally that are operating/in the process of implementing MCs. The CoLP interfaces with the MC staff and volunteers at the micro level to affect everyday operations (e.g., support activities) and, by extension, outcomes for people affected by dementia.

The WMCP is positioned at the meso level of the system because the Funder and Leader are the key stakeholders driving implementation. The former is responsible for drawing on the assets of people and places to create communities that promote health and happiness (‘placemaking’). Implementing the WMCP required support from health and care services operating in different sectors, demonstrated in the number of interactions between the WMCP and external stakeholders channelled through a select number of key contacts (e.g., NHS, Social Care, Public Health, third sector). These intermediaries supported scaling processes by mobilising resources, championing the WMCP vision, bridging communications, and increasing understanding (policies, systems, norms) between the WMCP Leader and health and care services ([18]).

Implementing the WMCP at the micro level involves PDS services and primary and secondary care. These systems work directly with people affected by dementia, providing information, advice, and referrals to MCs and acting as a key juncture between WMCP county-wide and local-level operations. The effects of failed market shaping in adult social care are visible in WMCP implementation ([74]). The supply of MCs was dominated by two Providers operating under a national charity for older adults, indicating a monopoly power and, thus, a concentration of power in WMCP implementation.

#### 3.2.3. Tailoring and Modification of Meeting Centres and WMCP During Implementation

Table 3 demonstrates what, why, and how individual MCs were tailored or implemented ‘adaptively’ based on local contexts ([60]).

The Leader initiated modifications to the WMCP through a ‘Task and Finish Group’, which intervened to drive improvements. The group met online monthly, including senior representatives of key systems, ICB, adult social care, and third-sector organisations. The modifications are summarised below.

The Leader delivered referrer workshops to educate professionals about the WMCP and how to refer people directly to MCs, connecting professionals to integrate services and strengthen the referral pathway. Workshops enabled professionals to share their experiences and feedback on the programme. Approximately 60 professionals attended two workshops, including Social Workers, Psychologists, and Occupational Therapists. Outputs included an MC referral form and a contact list of all the MCs to be publicly available. Senior leaders of MC providers positively reflected upon this modification:
“… you have [Leader] organized those meetings, you have got professionals together. I think that will make a huge difference down the line. I think referrals will start coming through more rapidly”(Chief Executive Officer of an organization providing MCs)


A Community Engagement Officer (EO) was employed 2.5 days a week for 12 months to raise awareness of the WMCP within public and third-sector organisations and to facilitate visits to MCs for people affected by dementia. The EO found limited professional awareness of the WMCP and accessibility issues, including membership costs and unattractive activity programmes. Thus, provider workshops and promotional videos were sub-components added to the WMCP. Furthermore, the EO role was found to be infeasible and ineffective due to the county-wide remit not being compatible with the part-time hours and objectives of becoming embedded in multiple local communities. The role was discontinued after 6 months, and instead, small amounts of funding were available for each Provider to engage the community locally.

“My role came from the … low numbers of people attending. Originally, the idea was that the EO would … get in touch with people in that kind of diagnosis pathway … and literally accompany people to the Meeting Centre for that first time… But the reality of how my role has panned out is quite different because I do not think it was particularly well conceived. It was based on experiences from MCs in other parts of the country where you had … one person who was … part of the local dementia care pathway who was able to support people coming to one Meeting Centre. Whereas … there are different organizations running the Meeting Centres … that community engagement needs to happen across the 10 Meeting Centres. One person doing 10 Meeting Centres in half a week. It is not feasible.”(Community Engagement Officer)

The EO facilitated a workshop for Providers to enable the adoption of EFs. The workshop explored the implementation challenges and opportunities, including uptake and professional engagement. While the focus was on reinforcing the intervention EFs, the opportunity for providers to interact was equally important:

“We are running this workshop next week … to try and support them to reflect on how well they are meeting the Essential Features… And actually, some of these people have never even met each other. And that is almost one of the most important bits of the workshop, is them talking to each other and sharing practice, sharing ideas, sharing what works for them and what does not work.”(Community Engagement Officer)

The Leader developed short (under 5 min) educational videos in collaboration with individual MCs to increase awareness of what an MC is and how it may benefit people and services. Videos targeted professional referrers (https://www.youtube.com/watch?v=DpzwtlduXho, accessed on 22 January 2025) and people affected by dementia (https://www.youtube.com/watch?v=5zQmQW2T3lU, accessed on 22 January 2025). Videos included the lived experiences of staff, volunteers, and people with dementia and footage captured at MCs.

### 3.3. How Well Was the Programme Implemented

#### 3.3.1. Development and Distribution of Resources

The programme implemented 10 MCs, with at least one in each of the Primary Care Networks and six districts in the county, which exceeded the expectations of the programme’s coverage. The WMCP focused on reducing health inequality by distributing MCs’ investments in “areas of greater disadvantage” (Integrated Care Board representative). We identified one MC implemented in a highly deprived neighbourhood; however, that MC was not sustained, raising equity concerns to the Funder and Public Health:

“… the feedback from that initial work … was that it just wasn’t working … whether there was a cost prohibitive element… But it was an area of concern that myself and public health colleagues had in terms of the relocation of that center to a different part of Worcester.”(Local Government representative)

The distribution of WMCP funding intended to promote a mix of third-sector providers. However, this goal was not achieved, and instead, the WMCP replicated existing market power through Providers operating under a national charity ‘dominating’ the supply chain. Subsequently, the market power could dictate the MC price, the support offered, and other outputs, resulting in fewer choices for people and professionals ([12]). The Chair of the Assessment Panel observed the following:

“I have seen it happen elsewhere where the bigger charities can soak up…the money because they have got the infrastructure…And if you want a quick win, that is often a good way to go. But if you actually want to make change…Then you need to make sure that no one party dominates…in an ideal world, we would have had ten different types of organizations delivering…Meeting Centres.”

Fidelity assessments completed by the Leader based on the EFs suggested all MCs were ‘working towards meeting the EFs’ during this research period. Areas of improvement related to the following: ‘Skilled and stable team’ (EF 4); ‘Leadership’ (EF 5); ‘Focus on people with dementia and unpaid carers’ (EF 6); ‘Programme of Activities’ (EF 7); and ‘Data Collection’ (EF 11) ([106]). Consistent fidelity issues caused by Providers reducing the intervention’s EFs (simplification) raise fundamental questions about their feasibility at scale. For example, developing economies of scale by sharing staff to reduce service delivery costs meant ‘non-essential’ components such as data collection and a stable staff team were overlooked.

#### 3.3.2. Meeting Centre Training and Spreading Learning/Lessons from WMCP

Providers should enrol staff and volunteers in the MC training course to ensure they have the knowledge and skills to replicate the EFs. Uptake was good, with around 73% of staff and volunteers having completed training, although for a period, there were some MCs that did not have trained personnel. This impacted the quality of support offered because most social care professionals, regardless of previous care experience, do not have the necessary skills to deliver complex multicomponent support programmes ([57]).

The online data collection support session aimed to ensure Providers understood how and why data were collected. However, an initial attempt to retrieve data found that health and wellbeing data were missing, and some MCs did not capture daily attendance. The Leader supported data collection at certain MCs to address these issues until university students from allied health courses were trained and collected data during their work experience placements.

#### 3.3.3. Monitoring and Evaluating the Programme to Create Feedback Loops to Reinforce/Modify WMCP

Monitoring and evaluation procedures were important feedback mechanisms that enabled knowledge to be captured and used by the Leader to reinforce or adapt scaling processes ([53]; [61]). The ‘Task and Finish Group’ fed into key services that engage with people affected by dementia to promote the programme’s aims. This had a variable impact due to existing health and social care pressures. MC effectiveness is highly dependent on the functioning of existing health and care pathways to ensure people utilise MCs as planned ([72], [69], [70]); therefore, services need to be sufficiently resourced and integrated to adopt WMCP activities and work collaboratively. For example, a Provider reported working with eleven Social Prescribers in two years, reducing the consistency of referrals. An explanatory sub-theme for this factor was the impact of the COVID-19 pandemic:

“COVID is the overriding factor here, in that the referral routes that used to be open are now closed…GP practices have been shut…and health professionals have been concentrating on vaccines…we know it has had a huge impact on people being diagnosed, even getting into the system in the first place, and also for people getting information about the services out there.”(Provider C, Chief Executive Officer)

The EO role, which was supplemented with funding for the organisations delivering MCs, provides further examples of using evaluative reasoning to modify the programme and demonstrate the complexity of using feedback loops in a meaningful way ([83]). Even when information was employed efficiently to modify the WMCP, it was sometimes counterproductive. For example, if stakeholders had assessed the scale of the issue, it would have been clear that a part-time EO could not have a meaningful impact on uptake county-wide. Similarly, while small pots of money may better serve local needs, the investment undervalued the problem, and the duration of the investment meant that any progress was unlikely to be sustained. Anecdotal evidence from Providers suggested this funding increased the number of referrals but not actual MC membership. Complex systems approaches may have been beneficial here to account for and address the internal and external complexity of scaling ([67]). There is a need to retrieve and respond to programme and external information, as these are not mutually exclusive, and to do so with the understanding that external components may also require scaling, as one interview participant pointed out:
“It is not as simple as assuming that you just need one thing…We have multiplied the number of Meeting Centres, but actually, there is some of those other sorts of strategic level stuff [that has] not been upscaled.”(Community Engagement Officer)


#### 3.3.4. The Key Role of Intermediaries and Infrastructure When Scaling-Out Meeting Centres

Intermediaries that mobilise resources and facilitate communication channels can support scaling processes ([18]). During the WMCP, their support was convened and orchestrated by the Leader. Participants described the Leader’s structural independence, strategic reach (e.g., Integrated Care System), and “kudos” (i.e., political capital) as key mechanisms to locating and involving intermediaries to drive systems and culture change, or, as one participant described the Leader “… got groups of people to do things” (Provider C, Chief Executive Officer). Examples of the convening capability of the Leader were efforts to address the strategic referral issue:

“…the really good thing … was the university taking very seriously the referral issue…that ended up being a kind of pincer movement with the university pushing that. Where, in one sense, the weight of the university can go and talk to that point in the NHS, at a slightly more senior level.”(Provider B, Senior Manager)

MCUK was an important infrastructure and asset when scaling MCs, saving the infrastructure to provide training, offer technical implementation support, and facilitate the CoLP. It provided the foundation for WMCP implementation and evaluation and, in doing so, reduced the workload and implementation costs to develop guidance and evaluation frameworks. The CoLP is a vehicle for Providers to utilise MCUK resources, which provide education to scale-out, accountably, the intervention principles and practices ([102]). Nevertheless, participation by Providers in the CoLP was poor, and overall, there was limited evidence that they used MCUK resources effectively. Although where there was evidence, this suggests it may improve service quality and collaboration between providers:

“… the main benefits have been through the wider network. We have probably learned more from the Scottish and the Welsh [Meeting] Centres. The online meeting … where people from different Meetings Centres across the country share knowledge … one of our trustees always goes… And she always comes back with ‘this is really interesting, we should try this’… So those have been hugely valuable.”(Provider B, Senior Management)

## 4. Discussion

This section discusses the implications of the key findings for research and practice, constraints upon these, and suggestions for further enquiry.

### 4.1. The Feasibility of the Adapted TIDieR Checklist to Report the Development of the Implementation of Scaling Interventions

The adapted checklist was a feasible framework to systematically and comprehensively report on the WMCP. Thus, it is an important contribution, given the absence of a reporting framework for complex scaling interventions ([40]). It ensured key programme information could be reported in a standardised format that incorporated items specific to controlled trials ([46]), complex health interventions ([24]), and scaling programmes ([40]). The checklist exemplified the qualities of a research design tool by involving key stakeholders to clarify the essential components and what worked (or did not) in practice ([1]). In addition, the process of explaining the antecedents, attributes, and intended consequences of a programme is indicative of theory development ([37]); therefore, this study has developed and modelled a descriptive theory of implementation that can aid the process of translating evidence into routine practice ([77]).

Further research should build on this theoretical contribution by evaluating how MCs work at different scales, for whom, and under what changing conditions ([61]; [98]). For example, understanding where intermediaries operate and their ability to support the programme seem to be important preconditions for intermediate outcomes in individual- and service-level effects ([18]). Such understanding can refine WMCP programme theory, for example, to identify implementation strategies that could target specific implementation outcomes ([91], [90], [89]) to improve outcomes for people affected by dementia ([9]).

### 4.2. Develop a Scaling Strategy Based on Optimal Scale

A scaling-out strategy for MCs and other community-based interventions should provide a realistic ‘vision of scale’ that can inform the development and continuation of programmes beyond the initial funding/investment ([53]). A focus on ‘maximal scale’ and the assumption that if something (an MC) is implemented, no matter how badly, then people will instinctively turn up, negatively impacts the reach and sustainability of the programme. Inadequate community engagement, situational analyses (i.e., scalability assessment), and economic forecasting resulted in limited evidence to indicate optimal scale ([22]; [53]) and how best to ‘sell’ the programme to key stakeholders ([51]). Consequently, the Funder’s investment was spread too thinly and exacerbated by low membership income from the limited uptake of MCs.

Therefore, most Providers simplified the EFs to improve feasibility and financial sustainability. This subtraction of complexity when scaling-out is common ([102]) and emphasises the importance of psychosocial interventions adaptive implementation at the local level ([60]). Nevertheless, simplifying the EFs altered the MCs from conception ([34]), suggesting the EFs are unfeasible when scaling-out. We argue that that is not true and that fidelity and, by extension, feasibility should be framed pragmatically as dynamic processes exhibiting varying levels of adherence and viability ([6]). Further research should re-examine the MC essential and non-essential features ([47]), providing a pragmatic measure of fidelity, including MCs’ ability to gain and apply power within/across allied services/systems ([67]). Such a measure could reduce the risk of prematurely discontinuing programmes, which could be beneficial in reducing waste.

In the case of the WMCP, the optimal scale was not determined, so it remains unclear whether MCs are a scalable solution to the PDS gap in the UK. Therefore, we recommend questioning, ‘What is the optimal scale for the WMCP?’ by evaluating the ability of the intervention to be scaled-out in the first instance ([66]; [118]). For example, considering trade-offs between the scale and scope of a programme (i.e., number and quality of MCs) and implementation outcomes such as equity, reach, and sustainability, as well as who will be the Funder, Leader, and intermediaries now and in the future ([53]). These findings should be considered in efforts to scale-out MCs nationally (such as those in Scotland) to help determine the optimal scale for MCs and whether these apply to other PDS interventions. There may be similarities and differences between what makes PDS programmes more or less scalable and, therefore, a source of competitive advantage in the UK’s marketised social care economy (e.g., cost-efficient, adaptable).

### 4.3. Work from the ‘Bottom-Up’ and Understand Equity of Impact

The process of scaling-out MCs involves centralised resources and management, as well as community engagement and ownership. Hence, research recommends combining ‘top-down’ and ‘bottom-up’ approaches ([79]). The WMCP exemplified a ‘top-down’ approach to service design and delivery, in the sense that senior leaders led on how the programme was developed and implemented, adopting fixed and predetermined funding, time scales, and objectives (“project approach”) ([53]). Top-down approaches can be time- and cost-efficient because they consolidate effort and decision-making, which, in theory, should also clarify accountabilities; however, in this case, this led to limited public and professional involvement that is not in line with co-production principles ([105]).

Short and fixed time frames are not conducive to solving complex issues at scale ([82]), particularly if the aspiration is to do so equitably ([53]; [61]). Therefore, Funders and Leaders should work to empower local actors and mitigate the project approach when scaling-out community-based dementia interventions. A move towards more co-operative community development and ownership models is advantageous. For example, place- and asset-based working are models advocated in adult social care for local authorities and communities to share power, respecting and building on local knowledge and expertise to improve health and wellbeing ([38]). There are recent examples of large-scale community-based support programmes for older people using an assets-based model successfully ([59]); we should build on these examples in psychosocial dementia care.

We believe the limited engagement with deprived communities exacerbated the challenges of embedding MCs in poorer neighbourhoods, where austerity policies in the UK have depleted the social infrastructure necessary to support the implementation of community programmes ([50]; [55]). It is unclear how the WMCP contributed to place-based inequities through equity of impact at scale. Therefore, we recommend completing an equity audit when scaling-out community-based dementia interventions. In the case of the WMCP, this should include how, through a focus on implementing MCs in “areas of greater disadvantage”, the programme addresses inequity in access, quality, and individual outcomes such as deprivation, ethnicity, age, and geography ([115]). Attending to this recommendation will demonstrate how programmes meet or intend to meet the expectations of the Equality Act 2010, specific national initiatives ([76]), and regional/local strategic needs (e.g., [116]).

### 4.4. How Findings Build on Existing Evidence on MC Implementation

In addition to the previous discussions, there are two key contributions this study makes that expand the understanding of the barriers and facilitators affecting the implementation of MCs internationally ([60]; [63], [62]; [108]).

This research shows that when scaling-out MCs, the needs and preferences of services/systems should be considered, in addition to people with dementia and unpaid carers. For example, the willingness and capacity of services/systems to create the desired outcomes through effective market shaping, awareness raising, and referral activities seems instrumental. These are likely being influenced by a combination of failures: market (insufficient competition), Government (insufficient funding), and third sector (insufficient skill and support) ([74]) that often leaves community-based dementia interventions at odds with their aims and what individuals and society need (blinded for peer review). Why the programme enabled a monopoly power and how this influenced the implementation outcomes would benefit further research; for example, sustainability, access, and equity are adult social care outcomes that *the market* influences ([74]).

[108] ([108]) found that the micro/individual-level facilitator was a leader who guided implementation. However, our findings show that when scaling-out MCs, the Leader operates at the meso/organisational level. The convening capability of the WMCP Leader brought together intermediaries, which were critical to implementing the programme, including modifications to address cultural and structural challenges. Their perceived impartiality, kudos, and strategic reach were key mechanisms for having the ability to convene key intermediaries. This is an important contribution to the literature on dementia intervention, which lacks evidence of the mechanisms affecting implementation outcomes ([56]; [113]; [119]). An output of this was the ‘Task and Finish Group’ to integrate and institutionalise the WMCP into key health and care services/systems (i.e., penetrate and adopt). Their activities aimed to reduce market competition and the duplication of effort, mobilise stakeholder resources more efficiently, and strengthen relationships. Such ways of working are essential to harness the power of integrated care system reforms in England ([8]; [44]).

## 5. Strengths and Limitations

This study addressed a common limitation of intervention reporting—a weak intervention description. Combining different reporting elements was a strength, aligning content to interventions generally, scaling programmes specifically, and the characteristics of intervention complexity ([24]; [40]; [46]). Furthermore, complementing document analysis ([13]) with different accounts from practice was an effective approach to reporting key programme information because it provided rich contextual detail, tracked emerging changes, and enabled the triangulation of evidence to increase the credibility of results ([92]). The findings provide a detailed account of the circumstances through which key stakeholders developed, implemented, and evaluated WMCP (i.e., context)—a core component when evaluating complex health interventions ([98]). Therefore, this study contributes much-needed evidence on developing and implementing scaling-out programmes in dementia care ([21]; [56]; [96]; [101]; [119]).

The limitations were the level of systematisation of how the document analysis method was applied. For example, not maintaining a detailed audit list of documents collected and analysed; therefore, some key documents may have been missing. The absence of an audit list also meant we did not report specific dates and other characteristics of the documents ([114]). Finally, the inclusivity of findings is a concern, as member-checking did not involve people with dementia, unpaid carers, or MC providers. The inclusion of different participants’ interview data offset this issue slightly as it broadened the voices included in this study, albeit indirectly.

## 6. Conclusions

This is the only study that reports an approach to scaling-out MCs in the UK, making an important contribution to their ongoing translation into mainstream policy and practice ([95]). The key findings were identifying WMCP components and modifications, the convening capability of the Leader, feedback loops to generate evidence for improvement, and the need for implementation strategies to utilise scaling principles and practices that include situational analysis and participatory approaches. These may help determine the optimal scale, integrate top-down and bottom-up perspectives, and develop shared outcomes, ownership, and accountability at the outset, therefore requiring less need to modify programmes that add to their overall complexity. To conclude, this research adds insight that can inform policymakers and practitioners considering MCs as a scalable solution to the gap in PDS, including factors that could inhibit or support the development and implementation of other community-based dementia interventions at a different scale.

## Figures and Tables

**Figure 1 behavsci-15-00670-f001:**
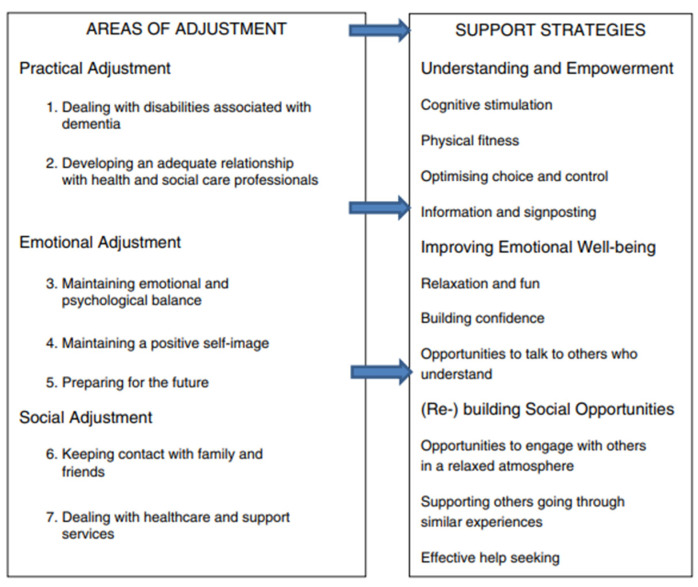
The Adjusting to Change model for people affected by dementia ([15]).

**Figure 2 behavsci-15-00670-f002:**
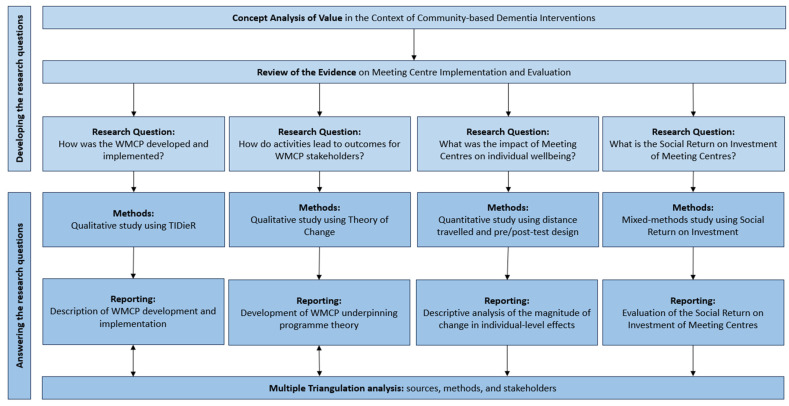
Flow chart showing the research preparation and delivery phases, including the formulation of questions, application of methods, and types of reporting.

**Figure 3 behavsci-15-00670-f003:**
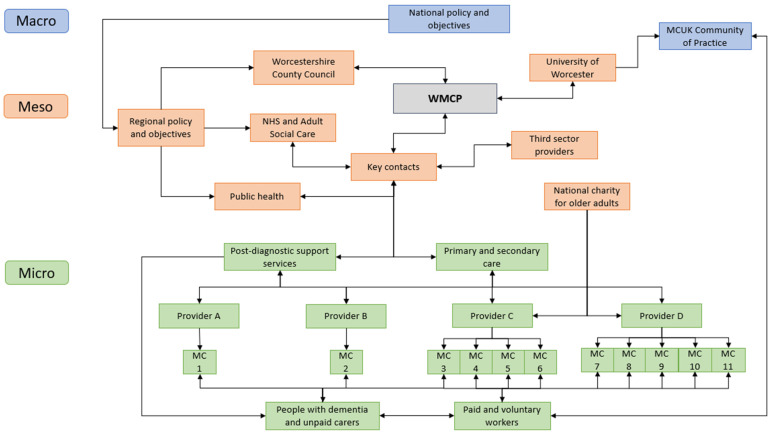
Overview of Worcestershire Meeting Centres Programme hierarchical implementation structure.

**Table 1 behavsci-15-00670-t001:** The framework to describe the Worcestershire Meeting Centres Programme (adapted from [24]; [40]; [47]).

Reporting Item	Description
Brief name *	Name and/or phrase that describes the intervention in straightforward terms.
Why */**/***	Any rationale, theory, or objectives of the programme, including key components, stage of intervention, and scalability.
Scaling strategy ***	The scaling strategy, including the specific use of scaling principles and/or implementation theory.
What *	What implementation materials and resources were used?
Who provided *	Key organisations and individuals and their technical name (e.g., Leader, Funder, Provider), expertise, and role in the programme.
How *	Implementation (in-person, remote) and format of programme modalities (individual, group-based).
Where *	Places and contexts where the intervention occurred.
When and how much *	Frequency and period the intervention(s) was delivered.
Tailoring *	Planned/unplanned personalisation and adaptation of the programme.
Modifications **	Unplanned changes made to the programme during implementation and any enabling factors.
How well */**/***	How well was the scaling strategy implemented?
	Was programme fidelity assessed, and what were the results?
	What was the impact of modifications on scaling outcomes?

Note: TIDieR item *, Adapted/extended TIDieR item **, SUCCEED item ***.

**Table 2 behavsci-15-00670-t002:** Description of Worcestershire Meeting Centres Programme components and sub-components.

Component	Sub-Components	Description
Develop and distribute resources	Promotion, publicity, and marketing	Spread awareness so that stakeholders adopt the programme
	An Assessment Panel was established to review and allocate funding awards	Reach consensus on application decisions
		Award funding to organisations to implement Meeting Centre(s)
		Review funding awards annually
	Meeting Centre fidelity, implementation, and evaluation	Guidebook on the Essential Features of a Meeting Centre
		Guidebook on Setting up and Running a Meeting Centre
		Data collection booklets and ‘how to’ guidance
		Technical support from the Leader to address emerging issues
Report on monitoring and evaluation	Monitoring, evaluating, and reporting objectives set by Funder	6-month report completed by Leader against ‘milestones and targets’
		Final programme evaluation of impact completed by Leader using routine data collected by MCs
		Assessment Panel meetings held periodically to monitor progress
	Monitoring, evaluating, and reporting objectives set by the Leader	Providers’ adherence to the Essential Features of a Meeting Centre is assessed and reported annually by the Leader and Providers
		Meeting Centre funding forecasts and overall financial viability are assessed and reported annually by Providers
		Routine data, including attendance, satisfaction, and wellbeing data, are collected and reported by Providers
Training and spreading learning	MCUK training course for staff and volunteers	Online short course focused on ‘Adjusting to Change’ theory and person-centred dementia care
	Collecting and sharing information with stakeholders (feedback loops)	Identify emerging issues and evidence to stimulate the progress of the programme
		Respond to emerging information to advance the aims of the programme
	MCUK Community of Learning and Practice [CoLP]	Opportunities to share and learn from current practice nationally
		Engage providers in adopting the principles and practices of MCs

**Table 3 behavsci-15-00670-t003:** What, why, and how was the intervention tailored to local fit?

What	Why	How
Targeted support for people living with Young-Onset Dementia	This population was not utilising the programme	Provider set up a Meeting Centre for people with Young-Onset Dementia Branding and publicity specifying support for people with Young-Onset Dementia (under 65 years) Increased focus on physical adjustment, including leisure and sports activities
Free membership	The membership cost is a barrier to uptake, and the cost-of-living crisis meant people had less purchasing power	Individual members subsidised by Providers for approximately 10 months
Number of days open per week	Low uptake and inflated venue costs meant it was financially unviable to increase weekly provision	In year three of the programme, one Meeting Centre opened 3 days per week, most opened 2 days, and some opened 1 day
Providing lunch for people with dementia	MCs opened during COVID-19 restrictions, and lunch could not be provided	MCs asked people with dementia to bring their lunch; this has continued after restrictions lifted
Having a stable staff team	Because there was a ‘monopoly power’ where ‘economies of scale’ were a focus (consolidate effort to reduce cost)	An MC Manager worked across several sites with no/limited support, as opposed to a manager at each with staff and volunteer support
Focus on unpaid carers as well as people living with dementia	Some providers were unaware of MCs’ dyadic focus and/or had reduced staff/volunteer capacity to engage unpaid carers meaningfully (e.g., facilitate peer support group)	Staff and volunteers encouraged unpaid carers to leave the Meeting Centre for care breaks Limited opportunities for unpaid carers to participate in relevant activities Most unpaid carers preferred not to attend
Types of activities/support offered	Person-centred practice means tailoring support to the interests, abilities, and circumstances of individuals	Offering a programme of activities that reflects individual Meeting Centre membership Offering ad hoc support to unpaid carers to address specific needs

## Data Availability

The datasets generated and analysed during the current study are available from the corresponding author upon reasonable request.

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
