# Peer review of "Development and Implementation of a Novel Approach to Scaling the Meeting Centre Intervention for People Living with Dementia and Their Unpaid Carers, Using an Adapted Version of the Template for Intervention Description and Replication (TIDieR) Checklist"

_behavsci, 2025, doi:10.3390/bs15050670_

Round 1

Reviewer 1 Report

Comments and Suggestions for Authors

The document discusses a pertinent topic using a coherent and methodologically rigorous approach. Nonetheless, other elements might be enhanced, including grammatical uniformity, clarity in articulating concepts, and rectification of typographical mistakes. It is recommended to improve conciseness in certain areas and refine the structure to facilitate a more seamless reading experience. Furthermore, increased focus on the practical ramifications of the findings could enhance the document's significance for stakeholders. Utilizing tables and visual aids may enhance the accessibility of the offered information.

The abstract is informative although may be simplified to enhance accessibility for a wider readership. Certain sentences, like "Complex interventions are often poorly described...", are excessively intricate and could be simplified into shorter, more comprehensible words. Terminology such as "scaling-out" and "scaling" must be used consistently to prevent confusion among readers unfamiliar with the differences. Moreover, the abstract should directly underscore the importance of the study's findings, illustrating how the intervention enhances dementia care.

Typographical errors exist, such as “post-diagnotsic,” which should be amended to “post-diagnostic.” Furthermore, “Implementioan outcomes” ought to be corrected to “implementation outcomes.” The application of commas in enumerations is uneven, and reorganizing these lists could improve clarity and readability.

The introduction offers a comprehensive history; however, it would improve with a more defined structure, incorporating subheadings to clarify the study's objectives, reasoning, and context. An improved, succinct, and captivating introduction would effectively engage the reader's interest from the first. Key concepts must to be defined earlier in the introduction to provide a robust foundation for comprehension. Moreover, an overabundance of citations in certain paragraphs can hinder the text's coherence; a more seamless integration of these references would enhance readability.

Complex words, such as "Drawing on the work of others, scaling is defined as a systematic effort...", could be divided into simpler components for enhanced understanding. The sentence "The disparity in implementation between successful pilots and routine adoption results in limited availability..." could be modified to enhance subject-verb agreement and clarity.

The methodology section is comprehensive and clearly articulated; nevertheless, the use of visual components such as tables or flowcharts to depict the research design would improve comprehension. The sequence of presenting qualitative and quantitative methodologies may be reevaluated for improved logical coherence. The justification for employing the modified TIDieR checklist requires further elaboration to clarify its selection and modification. A more detailed examination of the study's limitations in this section would be advantageous.

Repetitive statements, such as "This study was part of a pragmatic evaluation...", occur many times throughout the sections. To enhance readability, redundant expressions must be reduced. Moreover, maintaining uniformity in verb tense within the technique section (e.g., alternating between past and present tense) would improve coherence and clarity.

Results: Certain subsections in the results section exhibit an excessive degree of information, which may be more appropriately placed in appendices to preserve emphasis on principal findings. A more explicit differentiation between primary and secondary findings could enhance readers' comprehension of the study's fundamental outcomes. Furthermore, incorporating a concise analysis of the findings in this part would offer context and enhance the significance of the results. Uniformity in the vocabulary employed to describe program components is crucial to prevent ambiguity.

Typographical inconsistencies exist, including conflicting spelling of "scaling-out." Certain statements are excessively intricate and may be streamlined, for instance, "MC effectiveness is significantly reliant on the operation of current health and care pathways..." Streamlining these structures would improve reading and understanding.

The discussion part requires a more coherent structure, incorporating distinct subsections that address implications, constraints, and suggestions for further research. Certain assertions lack explicit citations and might be bolstered by incorporating supplementary evidence. Minimizing the reiteration of themes previously addressed in the findings section would enhance conciseness and clarity. Offering additional actionable ideas for policy and practice may augment the practical significance of the findings.

Errors like "In WMCP case" should be corrected to "In the case of WMCP." Sentence patterns must be examined to diminish passive voice utilization and enhance intelligibility. The coherence of thoughts could be enhanced by more effectively connecting sentences.

Comments on the Quality of English Language

The document's English language quality is satisfactory; yet, some places necessitate revision to improve clarity, readability, and professionalism. The following are detailed observations pertaining to grammar, sentence construction, lexicon, and stylistic elements:

The document exhibits grammatical mistakes, including subject-verb agreement discrepancies and inappropriate word selections. Phrases such as "the implementation outcomes pertaining to the inner and outer contexts that result in program modifications" could be reorganized for enhanced clarity by the use of commas and a rearrangement of the sentence to eliminate ambiguity.
The document has inconsistencies in verb tense, frequently alternating between present and past tense, perhaps causing confusion for the reader. Consistent application of either the past or present tense, contingent upon the context, would enhance readability.

Certain prepositions are misapplied or altogether absent, exemplified by "focus on program leaders who have convening capability to drive systems and culture change," which should be articulated as "a focus on program leaders who possess the capability to convene and drive systemic and cultural changes."
Numerous sentences are too lengthy and intricate, hindering the coherence of concepts. Decomposing lengthy phrases into shorter, more succinct statements would enhance understanding.
Certain words are inelegantly worded, impairing the document's readability. This study presents a fresh technique to scaling out Meeting Centres within this setting.
The prevalence of passive voice in the text renders the material impersonal and complicated. Utilizing the active voice in sentence construction, where feasible, would improve engagement and clarity.
The vocabulary employed is suitable for an academic context; yet, there are instances of duplication and repetition of crucial phrases, such "scaling" and "intervention." Incorporating synonyms and diverse sentence forms would enhance the text's dynamism and engagement.
Technical terminology is employed without sufficient clarification for readers lacking familiarity with the topic. Offering succinct definitions or contextual elucidations would enhance the clarity of essential concepts.
There are occurrences of erroneous or inconsistent terminology, exemplified by the varied references to "psychosocial intervention" throughout different sections. It is advisable to maintain consistency in vocabulary throughout the work.
Multiple typographical errors have been detected, including "implementioan" in place of "implementation" and "post-diagnotsic" instead of "post-diagnostic." A comprehensive proofreading procedure must be undertaken to detect and rectify these errors.
The application of commas is erratic, with several phrases missing essential commas to delineate clauses, thereby complicating readability. For example, incorporating commas in sentences containing numerous clauses might enhance readability.
Apostrophes are occasionally absent or incorrectly employed, especially in possessive constructions, such by "the program’s implementation outcomes" instead of "the programs implementation outcomes."
The document predominantly upholds an academic tone; nonetheless, there are sporadic informal phrases and colloquial idioms that ought to be substituted with more official alternatives.
Certain portions have a narrative style that diverges from the anticipated objective academic methodology. Adopting an objective tone by eliminating subjective statements would enhance the document's professionalism.
The document would improve with a more consistent application of transition words and phrases (e.g., "furthermore," "therefore," "however") to guarantee a logical flow of ideas.

Author Response

Please see word document.

Reviewer 2 Report

Comments and Suggestions for Authors

Thank you for this very detailed manuscript on an important issue. 

Some general comments: The syntax is at times very complex and difficult to follow or words seem to be missing, or ?? are in place of references. Please proof read. 

Methods: I do not understand what is meant be "epistemological flexibility, methodological comprehensiveness and operational practicality principles" (ll. 98-99). Please explain/ apply in relation to the presented study. 

The findings section is very lengthy, understandably, given the authors' aim to use TIDieR and a more detailed approach to intervention description. However, some paragraphs could be more concise to allow for better flow (e.g. 3.3). In section 3.10 it would be of interest to understand more precisely how to intervention was implemented as intended, and given that authors indicate that might have note been the discuss a more detailed discussion with regard to the underlying theory to explain why this might have been the case (contextual and theoretical reasons) as well as considerations of unintended consequences.

Additionally, section 4.4 seems oddly placed. Rather than a Q & A section and overall summary of recommendations for policy, practice and future research would be more suitable. 

Author Response

Please see word document.

Reviewer 3 Report

Comments and Suggestions for Authors

This article describes very clearly an important set of provisions established in Worcestershire for the support of people suffering dementia and their unpaid carers. Methodologically the account of how the network of Meeting Centres for these clients was established, structured and administered appears quite sound and it surveys carefully the strengths and weaknesses of the system, usefully recommending further research into both of these. It proposes that this model could be established nationally. A similar approach appears to be established in Scotland. However there is no explanation of how and why Worcestershire decided to establish and fund these Centres. Was it due to the presence in the county of the Centre for Dementia Studies at the University of Worcester which appears to have developed the idea of the Meeting Centres and where the research embodied in this article is based? Also how was a local council able to fund the Centres when it is well known that local authority funding has been severely cut since 2010, and local social services have been cut back as a result? Did they raise charitable or other funding? It is surprising too that there is no assessment of the effects of successful Meeting Centres upon the condition of dementia sufferers and the lives of their unpaid carers- surely the standards by which the value of the Centres must be assessed. And, since the researchers’ ambition is that similar schemes of good practice are desirable on a national level, especially in deprived areas, there is no indication of an intention to draw their findings to the attention of national policymakers to encourage them to extend their establishment. In my view this is an important article on a very important theme- the more so since the incidence of dementia is projected to grow in the UK in coming decades- but the article would be more valuable and have more impact if at least some of the omissions described above could be included in a revised version.

Author Response

Reviewer 3 comments

Authors response

In my view this is an important article on a very important theme- the more so since the incidence of dementia is projected to grow in the UK in coming decades- but the article would be more valuable and have more impact if at least some of the omissions described above could be included in a revised version.

We are proud that the reviewer feels this article is an important contribution and meets its aim of providing a very clear and detailed description of the support programme in Worcestershire.

We agree that the article could be improved by addressing ‘at least one’ of the reviewer's comments and would like to thank them for their time in reviewing our paper. In response, we have their omission, described below.

Methodologically the account of how the network of Meeting Centres for these clients was established, structured and administered appears quite sound and it surveys carefully the strengths and weaknesses of the system, usefully recommending further research into both of these. It proposes that this model could be established nationally. A similar approach appears to be established in Scotland. However there is no explanation of how and why Worcestershire decided to establish and fund these Centres. Was it due to the presence in the county of the Centre for Dementia Studies at the University of Worcester which appears to have developed the idea of the Meeting Centres and where the research embodied in this article is based? Also how was a local council able to fund the Centres when it is well known that local authority funding has been severely cut since 2010, and local social services have been cut back as a result? Did they raise charitable or other funding?

Given that the paper aims to adequately describe the programme for replication, we felt the most important omission to address was how and why the programme was funded. We have now included details on the points alluded to by the reviewer:

Lines 276-283: added funding mechanism for the council to finance WMCP. Included MCUK being established in 2018 as a factor that contributed to WMCP being funded.

It is surprising too that there is no assessment of the effects of successful Meeting Centres upon the condition of dementia sufferers and the lives of their unpaid carers- surely the standards by which the value of the Centres must be assessed.

We agree that the impact on individuals is an important consideration when reporting on complex interventions, however, it was not within the scope of this paper and will be reported separately as part of a cost-benefit analysis. The focus of this paper was on how and why a countywide support program was developed and implemented. This paper adds critical context for the outcomes paper because it shows the preconditions (inner and outer contexts) for individual- and service-level effects.

And, since the researchers’ ambition is that similar schemes of good practice are desirable on a national level, especially in deprived areas, there is no indication of an intention to draw their findings to the attention of national policymakers to encourage them to extend their establishment.

This research has been subject to standard dissemination practices (papers, presentations), and I have consulted on the findings with key stakeholders in Scotland and England; however, I have not yet found a way to speak with national-level policy makers. For these reasons, I have chosen to conclude that ‘this research adds insight that can inform policymakers and practitioners considering MCs as a scalable solution to the gap in PDS, including factors that could inhibit or support the development and implementation of other community-based dementia interventions at a different scale.’. (Pages 765-769)

Round 2

Reviewer 3 Report

Comments and Suggestions for Authors

The authors have responded adequately to my comments and the revised article is fit for publication